# Mosquito Population Dynamics and Blood Host Associations in Two Types of Urban Greenspaces in Coastal Florida

**DOI:** 10.3390/insects16030233

**Published:** 2025-02-20

**Authors:** Yasmin V. Ortiz, Simon A. Casas, Minh N. D. Tran, Emily G. Decker, Ivana Saborit, Hao N. Le, Eric P. Caragata, Lawrence E. Reeves, Panpim Thongsripong

**Affiliations:** 1Florida Medical Entomology Laboratory, Institute of Food and Agricultural Sciences, University of Florida, Vero Beach, FL 32962, USA; ortiz.yasmin@ufl.edu (Y.V.O.);; 2Entomology and Nematology Department, Institute of Food and Agricultural Sciences, University of Florida, Gainesville, FL 32611, USA; 3Department of Biological Sciences, Indian River State College, Fort Pierce, FL 34981, USA

**Keywords:** Culicidae, bite nuisance, blood meal analysis, community diversity, disease risk, invasive species, mosquito-borne disease, mosquito surveillance, urban parks, urban planning

## Abstract

Mosquitoes are known carriers of disease-causing pathogens that affect humans, and their populations can vary depending on landscape types. This study aimed to conduct mosquito surveillance in greenspaces within Vero Beach, Florida, and to understand how different types of greenspaces, specifically residential parks and conservation areas, affect mosquito numbers and their feeding patterns. By collecting adult mosquitoes from four greenspaces, we found that wetland-associated mosquitoes were more common in coastal conservation areas. Meanwhile, species known to carry human pathogens were more commonly found in residential parks. Blood meals from collected mosquitoes were analyzed to determine the animals they had fed on. While some mosquito species primarily fed on birds, others primarily fed on mammals. However, these patterns did not vary significantly between greenspaces, suggesting stable feeding behaviors across the greenspace types. This research highlights that not all greenspaces provide habitats for the same mosquito species. Further research can provide insights into how various green areas near human dwellings can shape mosquito communities, which can aid urban planning decisions in Florida.

## 1. Introduction

Mosquitoes serve as vectors for various pathogens, posing significant public health risks by transmitting diseases such as West Nile encephalitis and dengue fever [1,2]. Understanding mosquito interactions with landscapes and vertebrate hosts can reveal insights into risks of disease transmission [3,4,5]. This information can also aid understanding of how anthropogenic landscape changes and zoning patterns might impact public health.

Urbanization has been shown to significantly impact mosquito communities [6]. Within the urban environment, greenspaces can influence the composition of mosquito assemblages [7,8,9]. We define urban greenspaces as areas of vegetation located within or surrounding human settlements where its existence requires human involvement like planning, maintenance, or conservation effort [10]. This definition contrasts with a broader concept which defines a greenspace as any “nature” or natural area in general [10].

Even though urban greenspaces offer multiple health benefits to urban residents [11], there is concern about unintended negative effects, such as an increase in mosquito bites that are either a nuisance or pose disease transmission risks [7,12,13]. A recent metanalysis, however, found no significant difference in the mean abundance of three mosquito genera (*Aedes*, *Culex*, and *Anopheles*) or vector species, including *Aedes albopictus* Skuse and the *Culex pipiens* Linnaeus complex species, when comparing blue–green spaces (i.e., areas with vegetation and/or water) to non-greened urban spaces [8]. On the contrary, the abundance of *Aedes aegypti* L., an important vector species, was significantly higher in traditional urban areas void of blue–green space [8]. Notably, the study highlighted the variety of blue–green spaces and emphasized the need for further research to examine how different types of greenspaces uniquely influence urban mosquito communities. Such investigations are essential to better understand the ecological dynamics of greenspaces and their potential role in shaping mosquito populations and associated disease risks in urban areas.

In Florida, where locally acquired cases of mosquito-borne diseases such as West Nile encephalitis and dengue have been reported [14,15,16], few studies have directly compared mosquito communities across greenspace types. Most research has focused on identifying associations between land cover or habitat types with specific mosquito species. For example, Giordano et al. investigated mosquito community composition among various natural habitat types in eastern Florida and found greater species richness in mixed hardwood–coniferous and hydric hammock habitats compared to the scrubby pine habitat [17]. Another study compared urban farms in Miami-Dade County and observed differences in mosquito community composition and diversity between the two farm sites [18]. Other research has identified land cover associations with key vector species such as *Ae. aegypti*, *Ae. albopictus*, *Culex nigripalpus* Theobald, and *Culex quinquefasciatus* Say [19,20,21,22]. These findings highlight the influence of habitat type on mosquito assemblages, yet a direct comparison of mosquito communities across greenspace types remains limited.

Vector-borne disease transmission relies on interactions between vertebrate hosts and blood-feeding mosquitoes. Thus, understanding host–vector contact patterns through blood-feeding, in addition to vector abundance, is crucial for assessing disease risk. Mosquito species vary in their degree of host association: some are strongly associated with specific host species, while others display weaker associations, resulting in a wider host range [23,24,25,26]. By characterizing mosquito–host associations across different greenspace types, we can better understand how urban landscapes shape mosquito–host dynamics, which in turn may influence the likelihood of pathogen transmission. Identifying whether certain greenspaces facilitate greater contact between vectors and reservoir hosts or vectors and humans is essential for evaluating potential disease hotspots and informing urban planning strategies that mitigate public health risks.

This study aims to characterize and compare mosquito assemblages and their host associations from two greenspace types within a coastal city of Florida: residential greenspaces and conservation greenspaces. Specifically, we report surveillance data and assess whether greenspace type affects the relative abundance of key vector species, including *Cx. nigripalpus*, *Cx. quinquefasciatus*, *Ae. aegypti*, and *Ae. albopictus*, as well as an important nuisance species, *Aedes taeniorhynchus* Wiedemann. Through blood meal analysis, we also examined and compared mosquito–host interactions between these two greenspace types. This study provides insight into how urban greenspace types may influence the risk of mosquito-borne disease transmission. As Florida continues to expand and integrate greenspaces into coastal city planning, a nuanced understanding of how these spaces affect mosquito ecology and host interactions is crucial.

## 2. Materials and Methods

### 2.1. Study Sites

Mosquitoes were collected from four study sites in the city of Vero Beach, located in Indian River County on Florida’s Atlantic coast (Figure 1). Two greenspace types, residential and conservation greenspaces, were represented at the study sites. We defined residential greenspaces as public or private recreational areas within a town or city, typically surrounded by residential or commercial zones. In contrast, the conservation greenspaces were larger and specifically designated to preserve natural habitats.

In this study, the two residential greenspaces were Charles Park, a zoned public park, and the Loyal Order of Moose Lodge #1822 (hereafter referred to as Moose Lodge), a private recreational park, zoned as a general commercial district. These sites are located further inland relative to the conservation greenspace sites and are situated away from the Indian River Lagoon. Charles Park consisted of walking trails through pine flatwoods habitat, surrounded by residential buildings. Moose Lodge included areas near a small pond and pine flatwood habitat surrounded by commercial and residential developments.

The two conservation greenspaces were the Oslo Riverfront Conservation Area and the Indian River Lagoon Greenway. Both greenspaces were zoned as public lands conservation districts by Indian River County Planning & Development Services Department (https://indianriver.gov/services/community_development/index.php, accessed on 14 August 2024). They are adjacent to the Indian River Lagoon, a diverse estuarine ecosystem featuring habitats such as coastal hammocks, scrubby flatwoods, mangrove swamps, and impounded estuarine wetlands [27].

All study sites permitted active human use during the study period. All study site maps were created in R version 4.3.2 and RStudio 2024.04.0+735 using OpenStreetMap data, available as open data under the Open Data Commons Open Database License (OdbL; openstreetmap.org).

### 2.2. Mosquito Collection

Adult mosquito collection took place over 26 sampling weeks from February to December 2023. This timeline included the hot and rainy season spanning from late May to early October and the cool and dry season spanning from December through February [28]. The collection schedule consisted of four consecutive weeks of sampling, followed by three weeks without sampling. During each sampling week, all four sites were visited, with one conservation greenspace and one residential greenspace sampled on the same day. Sampling times and days were systematically alternated to ensure each site was sampled at alternating times and days throughout this study.

At each site, two types of CO_2_-baited traps were used: modified CDC miniature light traps (John W. Hock Company, Gainesville, FL, USA) and BG-Pro traps (BioGents^®^, Regensburg, Germany). Due to inconsistent light operation, the lights were removed from both traps. Human scent lures were not used and only a CO_2_ attractant was included. The CO_2_ attractant was released from approximately 680 g of dry ice placed in plastic Igloo^®^ coolers with flip-lids. These containers were positioned directly above or next to the trap entry points. The CO_2_-based trapping sessions began between 3:30 and 8:30 p.m. and ended between 7:15 and 9:00 a.m. the following morning, resulting in an average operation time of approximately 15.8 h (SD = 1.28). Only two CO_2_-baited traps were used per sampling site and were hung at least 0.32 km from each other in opposing directions. The CO_2_-baited traps were hung at least 0.61 m above the ground and no more than 1.22 m off the ground. The CO_2_-baited traps at each site were near or directly above brush/vegetation, as well as small collections of water (e.g., partially filled ditches, water-filled containers, etc.) when available. The distance between the CO_2_-baited traps and nearby vegetation or residential buildings was not measured for each trapping session. Additionally, large-diameter aspirators (LDAs), constructed based on designs from Sloyer et al. [29], were used by two operators for a total of 30 min (15 min per operator) per site per visit. These LDA-based trapping sessions were conducted between 7:00 and 9:00 a.m. on the mornings that CO_2_-baited trapping sessions ended.

### 2.3. Mosquito Sample Identification, Midgut Dissection, and DNA Extraction

Trap contents were transported on ice to the Florida Medical Entomology Laboratory in Vero Beach (approximately eight km from the study sites). Only female mosquitoes were counted and assigned to species using mosquito identification keys [30,31]. If sample damage prevented species identification, mosquitoes were categorized as unknown. Mosquitoes visually identified with full or partially blood-engorged abdomens were set aside for blood meal analysis. Midgut dissections were performed using sterilized equipment (e.g., forceps and glass microscope slides). The equipment cleaning procedure involved a wash with freshly prepared 10% bleach solution, followed by two washes with molecular biology-grade HyClone™ water (Cytiva, Wilmington, DE, USA). Mosquitoes were also surface sterilized using this protocol immediately before dissection. The midgut dissections were conducted on ice using forceps to gently extract the midguts. The DNA extraction from dissected midguts used the NucleoSpin Tissue Prep Mini Kit (Macherey Nagel, Dueren, Germany), following the manufacturer protocol. DNA quantity and quality were assessed using a Qubit 1X dsDNA HS Assay kit on a Qubit 4 Fluorometer (Thermo Scientific, Waltham, MA, USA) and Nanodrop Spectrophotometer (Thermo Scientific, Waltham, MA, USA).

### 2.4. Blood Meal Analysis

The DNA barcoding regions of the *cytochrome c oxidase subunit I* (*COI*) gene were amplified following the protocols described by Reeves et al. [32]. All DNA samples were initially amplified with the Mod_RepCOI_F + VertCOI_7216_R (244 bp) primer combination [32]. Successful amplification attempts were made using the Mod_RepCOI_F + Mod_RepCOI_R (664 bp) and VertCOI_7194_F + Mod_RepCOI_R (395 bp) primer combinations for 16 out of 245 samples that did not initially amplify (Appendix A). Of the final set of successfully sequenced samples, the majority (95) were from the 244 bp primer set, with one sample from the 395 bp primer set and five from the 664 bp primer set (Appendix A). During each PCR reaction, a no-template control (NTC) was included to monitor for possible contamination during laboratory procedures. The PCR products were assessed visually through gel electrophoresis, and successfully amplified products were sent for Sanger sequencing (Eurofins Genomics LLC, Louisville, KY, USA).

Sequence chromatograms were manually inspected and edited for quality using Geneious Prime^®^ (v.2023.1, Biomatters Ltd., Auckland, New Zealand). Sequences shorter than 50 bp after editing were excluded from further analysis. The final edited sequences were then submitted to the National Center for Biotechnology Information (NCBI) Nucleotide Basic Local Alignment Search Tool (BLASTn suite), with host type-level (e.g., mammals, birds, and reptiles) and species-level taxonomic identity assigned to sequences showing ≥80%, and ≥97% identity, respectively, to entries in the NCBI database.

### 2.5. Data Analysis

All data analyses were conducted in R version 4.3.2 [33] and Rstudio version 2024.04.0+735 [34]. A full table listing female mosquito abundances by species for each study site is provided in Appendix A. Mosquito assemblage diversity and sampling effort between sites were assessed using rarefaction curves, created using the iNEXT package [35,36]. Measures of alpha diversity, including species richness, the Shannon index, the Simpson index, Chao1, and the ACE index, were implemented using the vegan package [37]. Estimates of species richness and similarity per site were also calculated using the Jaccard Index.

To evaluate the relationship between weather variables, greenspace type, and the number of female mosquitoes, generalized linear models were used. Daily meteorological data were obtained from the Vero Beach Regional Airport (WBAN:12843 station, located at Lat 27.6553°, Lon 80.41425°, source: Local Climatological Data from the National Oceanic and Atmospheric Administration). Statistical analyses were performed to understand the impact, if any, of the average daily temperature (°F) on the collection date, cumulative precipitation over the three weeks prior to collection, and greenspace type on the count of mosquito species of interest collected per day at each site. The mosquito species of interest were *Ae. aegypti*, *Ae. albopictus*, *Ae. taeniorhynchus*, *Cx. quinquefasciatus*, and *Cx. nigripalpus*, for their roles as vectors or nuisance species.

This analysis was developed based on hypotheses drawn from the existing literature. Previous studies have shown that rising temperatures during warmer seasons are positively correlated with mosquito counts in traps [38,39,40,41,42], and that rainfall in the weeks preceding sampling also influences mosquito populations [42,43,44,45]. A three-week cumulative rainfall period was chosen based on prior studies [46,47,48], and because this interval generally aligns with the typical time from egg hatching to adult emergence and host-seeking behavior. Additionally, the initial exploratory analysis of correlations across various lag times (one, two, and three weeks) indicated that the three-week cumulative rainfall period had the strongest correlation with total female mosquito counts (Appendix A).

For each of the five species, generalized linear models were fitted using the MASS package [49]. Records with missing values in any predictor or outcome variable were excluded. Independent variables included the average temperature on the day of trapping (Appendix A), cumulative rainfall over the preceding three weeks, and greenspace type. Model diagnostics were performed using the DHARMa package [50] to assess residual patterns, dispersion, outliers, and potential zero inflation. Residuals were plotted against each independent variable to evaluate homogeneity and model fit. If necessary, mosquito counts were log-transformed to improve model fit. Temporal autocorrelation of residuals was assessed with autocorrelation (ACF) plots. If autocorrelation was detected, an autoregressive term (the count from the previous week) was added to the model to address this issue. The variance inflation factor was calculated using the vif function from the car package [51] to check for multicollinearity among predictors. McFadden’s Pseudo R^2^ was calculated to estimate model explanatory power.

For each species of interest, an additional model incorporating a quadratic term for temperature was evaluated to test for a non-linear relationship between temperature and mosquito count. Model fit was compared using the Akaike information criterion (AIC), with a lower AIC indicating a better fit. A likelihood ratio test was performed using the anova() function with a chi-square test (test = “Chisq”) to determine if adding the quadratic term for temperature significantly improved model fit.

To characterize blood meal types found in mosquito samples across different land uses, the relative abundance of host types and species identified through blood meal analysis were tabulated for each mosquito species by land use types and month of collection. The resulting data were presented visually using heatmaps (ggplot2 package) [52] and the parallel set plot (ggalluvial package) [53]. Fisher’s exact test was used to determine if *Ae. taeniorhynchus* and *Cx. nigripalpus* were significantly associated with mammalian and avian hosts, respectively, compared to other mosquito species. These two species had sufficient numbers of blood-fed individuals with identified host species allowing for statistical comparisons. To assess the potential influence of habitat type on host association, a logistic regression model with an interaction term between mosquito species and greenspace type was applied. This model analyzed whether associations with mammalian or avian blood meals differed by habitat for *Ae. taeniorhynchus* and *Cx. nigripalpus*, respectively. Additionally, rarefaction curves were generated to show the expected number of host species identified as a function of blood-fed sample size for *Ae. taeniorhynchus*, *Cx. nigripalpus*, and *Culex interrogator* Dyar and Knab, which had at least seven blood meal hosts identified to the species level.

## 3. Results

### 3.1. Mosquito Abundance

We collected a total of 19,181 female mosquitoes across all study sites. Of these, 18,550 individuals were successfully assigned to species. For 631 individuals, damage to the specimen prevented species identification. Of those successfully assigned to species, significantly more mosquitoes were collected from the conservation greenspaces (N = 14,184) than the residential spaces (N = 4366; χ^2^ = 5196.4, df = 1, *p*-value < 0.001). Among the study sites, the highest number of female mosquitoes was collected from the Indian River Lagoon Greenway (N = 10,986), followed by the Oslo Riverfront Conservation Area (N = 3198), Charles Park (N = 2580), and Moose Lodge (N = 1786). A chi-square goodness-of-fit test indicated that the observed mosquito counts at the four sites differed significantly from what would be expected under a uniform distribution (χ^2^ = 5196.4, df = 3, *p* < 0.001).

The highest number of female mosquitoes collected in a single trapping session was 4366 female mosquitoes collected on 31 May 2023, from a CO_2_-baited trap in the Indian River Lagoon Greenway (Figure 2a). Across all trapping sessions, 5.3% of CO_2_-baited trap sessions and 2.9% of LDA sessions yielded no mosquitoes. In total, the LDA collected 4907 mosquitoes, of which 9.39% were blood fed. The CO_2_-baited trap captured 13,643 mosquitoes, of which 0.53% were blood fed.

### 3.2. Mosquito Species

Across all sites, a total of 32 species, including the *Anopheles crucians* Wiedemann complex treated here as a single species, were identified (Appendix A). The most abundant mosquito species collected overall was *Ae. taeniorhynchus* (N = 8211). This was followed by *Cx. nigripalpus* (N = 7580) and *Culex pilosus* Dyar and Knab (N = 531). Other commonly found species are shown in Figure 2c. In the residential greenspaces, *Cx. nigripalpus* was the most frequently collected species, accounting for 68% of mosquitoes at both Charles Park and Moose Lodge. In contrast, *Ae. taeniorhynchus* dominated at the Indian River Lagoon Greenway, representing 62% of all mosquitoes identified from this conservation greenspace. The most common species collected from the Oslo Riverfront Conservation Area, another conservation greenspace, was also *Ae. taeniorhynchus* (43%), although *Cx. nigripalpus* was nearly as abundant (35%).

In addition to *Ae. taeniorhynchus* and *Cx. nigripalpus*, important vector or nuisance species identified in the study sites were *Cx. quinquefasciatus* (N = 91), *Ae. albopictus* (N = 49), and *Ae. aegypti* (N = 91; Figure 3). Also of note is a female *Aedes scapularis* Rondani collected in the Indian River Lagoon Greenway on 19 July 2023 which represented the first record of this non-native species in Indian River County [54]. Other non-native species found included *Aedes pertinax* Grabham, *Culex coronator* Dyar and Knab, and *Cx. interrogator*, which were collected from all sites.

### 3.3. Mosquito Diversity

A rarefaction plot, showing the expected number of species for a given sample size, is shown in Figure 2b. The 95% confidence intervals of the rarefaction curves for all sites overlapped, suggesting similar alpha diversity among sites. The species richness, Chao1, ACE, Simpson, Shannon, and Pielou indices are listed in Table 1. Species richness was highest at Charles Park, with 29 species identified, while the other sites each recorded 28 mosquito species. The Chao1 and ACE index indicated that the highest diversity occurred in the Indian River Lagoon Greenway, a conservation greenspace (31.33), whereas Shannon and Simpson diversity indices indicated that the highest diversity occurred in the Oslo Riverfront Conservation Area, the other conservation greenspace. The Pielou evenness index indicated that the most and least even mosquito communities were found in the Indian River Lagoon Greenway and the Oslo Riverfront Conservation Area, respectively.

The Jaccard similarity index measures how many species two communities share relative to the total number of species present in both. A value close to 1 indicates a high overlap in species composition. The Jaccard similarity of species composition between the two residential greenspaces (i.e., Charles Park and Moose Lodge) was calculated to be 0.9, while the Jaccard similarity between the two conservation land use sites (i.e., the Indian River Lagoon Greenway and the Oslo Riverfront Conservation Area) was 0.8. Comparing the mosquito species composition between the two greenspace types resulted in a Jaccard similarity of 1, indicating that all mosquito species found in residential greenspaces were also present in conservation sites, and vice versa.

### 3.4. Variables Affecting Mosquito Count

The relationships between independent variables (average temperature on the day of capture, the cumulative rainfall three weeks prior, and greenspace type) and the number of female mosquitoes collected per trap for *Ae. taeniorhynchus*, *Ae. aegypti*, *Ae. albopictus*, *Cx. nigripalpus*, and *Cx. quinquefasciatus* were assessed using generalized linear models with a negative binomial distribution (Appendix A).

The models for *Ae. aegypti*, *Ae. albopictus*, and *Cx. quinquefasciatus* (Appendix A, Table 2) counts included cumulative rainfall three weeks prior to the collection, average temperature on the day of collection (as a linear term), and greenspace type as independent variables. According to these models, temperature on the collection day had positive and significant associations with the counts of *Ae. aegypti* (*p* < 0.001) and *Ae. albopictus* (*p* = 0.006) but not *Cx. quinquefasciatus* (*p* = 0.447). In addition, greenspace types have significant association with both *Ae. aegypti* (*p* < 0.001) and *Ae. albopictus* (*p* = 0.019) but not *Cx. quinquefasciatus* (*p* = 0.100). Specifically, residential greenspaces were associated with higher counts of both *Ae. aegypti* and *Ae. albopictus* compared to conservation greenspaces. Finally, accumulated rainfall had a positive and significant association with the count of *Cx. quinquefasciatus* (*p* = 0.021) but not with *Ae. aegypti* (*p* = 0.703) and *Ae. albopictus* (*p* = 0.251).

The models for *Ae. taeniorhynchus* and *Cx. nigripalpus* counts included cumulative rainfall three weeks prior to collection, a quadratic term for average temperature on the day of collection, greenspace type, and an autoregressive term (count from previous week). According to the models, cumulative rainfall over the previous three weeks had a significant positive association with *Cx. nigripalpus* (*p* < 0.001) but no association with *Ae. taeniorhynchus* (*p* = 0.982). The first component of the quadratic term for temperature had a positive association with both species, but the relationship was significant only for *Ae. taeniorhynchus* (*p* < 0.001), suggesting the abundance of this species increased as the temperature increased. The second component had a significant negative association with both *Ae. taeniorhynchus* (*p* = 0.043) and *Cx. nigripalpus* (*p* = 0.003), suggesting that mosquito abundance initially increased with temperature but decreased once an optimal temperature is exceeded. The effect of greenspace type was significant only for *Ae. taeniorhynchus* (*p* < 0.001), where its count was significantly lower in the residential greenspace compared to the conservation greenspace. However, there was no significant effect of greenspace type on *Cx. nigripalpus* (*p* = 0.064).

The performance of the models was assessed using pseudo-R^2^, which provided indication of how well each model explained the variability in mosquito counts (Table 2). The models had pseudo-R^2^ values ranging from 0.13 to 0.51, indicating varying levels of explanatory power. The model for *Cx. quinquefasciatus* had the lowest pseudo-R^2^ (0.13), which suggested a limited ability to explain the variability in mosquito counts, likely due to fewer or less informative explaining variables. The model for *Ae. taeniorhynchus* had the highest pseudo-R^2^ (0.51) indicating the model could capture a substantial portion of the variability.

### 3.5. Blood-Fed Mosquitoes

A total of 533 mosquitoes were visually identified as blood fed. Among these, 172 samples came from conservation greenspaces (99 from the Indian River Lagoon Greenway and 73 from the Oslo Riverfront Conservation Area), while 361 were from residential greenspaces (204 from Charles Park and 157 from Moose Lodge), representing 1% and 8% of the total mosquitoes in each greenspace category, respectively. The mosquito species with the highest number of blood-fed individuals was *Cx. nigripalpus* (N = 212), followed by *Ae. taeniorhynchus* (N = 74).

Out of the total blood-fed samples, 398 mosquito midguts were successfully dissected. DNA from 169 midguts produced amplicons and high-quality *COI* sequences. Following quality filtering, *COI* sequences from 101 samples were retained for host species identification (Appendix A). These samples included 46 from Charles Park, 11 for Moose Lodge, 26 for the Indian River Lagoon Greenway, and 18 for the Oslo Riverfront Conservation Area. Host blood identification was conducted for *Cx. nigripalpus* (N = 48), *Ae. taeniorhynchus* (N = 23), *Cx. interrogator* (N = 7), *Cx. quinquefasciatus* (N = 5), *Aedes infirmatus* Dyar and Knab (N = 3), *Culex erraticus* Dyar and Knab (N = 3), *Culex salinarius* Coquillett (N = 2), *Culex iolambdis* Dyar (N = 1), *Uranotaenia lowii* Theobald (N = 1), and unknown species (N = 8). A large proportion (84%) of blood-fed mosquitoes in residential sites were of the *Culex* species, compared to 41% in conservation greenspaces. In conservation greenspaces, 48% of blood meals came from *Aedes* spp., with a majority being *Ae. taeniorhynchus*.

### 3.6. Association Between Mosquitoes and Their Hosts

Identified vertebrate hosts included amphibians (N = 1), reptiles (N = 10), birds (N = 39), and mammals (N = 51). Interactions between mosquitoes and vertebrate host types, categorized by sites, are shown using parallel set diagrams (Figure 4). For *Ae. taeniorhynchus*, a large proportion of blood meals overall were from mammals (*n* = 20, 87%). Using Fisher’s exact test, *Ae. taeniorhynchus* was significantly more likely to be found with mammalian blood meal compared to all other mosquito species (*p* < 0.001). The odds of *Ae. taeniorhynchus* found with mammalian blood meals were approximately 10 times higher than those of other mosquito species (Odds Ratio = 10.35, 95% CI: 2.70–59.57). For *Cx. nigripalpus*, hosts include birds (N = 24), mammals (N = 17), and reptiles (N = 7). Fisher’s exact test showed that *Cx. nigripalpus* was significantly more likely to feed on birds than other mosquito species (*p* = 0.018, Odds Ratio= 3.05, 95% CI: 1.18–8.32).

Logistic regression with an interaction term was used to determine associations between mosquito species and greenspace type with feeding on a particular host type. The associations of *Ae. taeniorhynchus* with mammalian blood meals and of *Cx. nigripalpus* with avian blood meals did not differ significantly between the two greenspace types (*p* = 0.164 and *p* = 0.266, respectively; see Appendix A for detailed results).

All three blood meals of *Ae. infirmatus* were derived from mammals. Other *Culex* species fed on birds, mammals, and reptiles. Reptiles were fed upon exclusively by *Culex* spp. (*Cx. nigripalpus*, *Cx. quinquefasciatus*, *Cx. interrogator*, and *Cx. iolambdis*). Only one amphibian blood meal was identified, and it was found in *Ur. lowii*.

When analyzing blood meal analysis results at the species level (Figure 5a), the most common vertebrate host species found overall was *Zenaida macroura* L. (mourning dove; N = 10), followed by *Gallus gallus* L. (chicken; N = 7) and *Cardinalis cardinalis* L. (northern cardinal; N = 6). Combining all blood-fed mosquito species, the number of host species identified in Charles Park was highest with 15 host species identified. This was followed by the Indian River Lagoon Greenway (nine species), the Oslo Riverfront Conservation Area (seven species), and Moose Lodge (five species).

*Culex nigripalpus*, *Cx. interrogator*, and *Ae. taeniorhynchus* were the three species for which the blood hosts of at least seven blood-fed mosquitoes were successfully assigned to species. *Culex nigripalpus* fed on a variety of host species, with blood obtained from 16 vertebrate species identified from 29 blood-fed mosquito individuals. Similarly, *Cx. interrogator* was found to have fed on a variety of host species, with blood from seven vertebrate species identified in seven blood-fed individuals. Twelve blood-fed *Ae. taeniorhynchus* took blood meals from seven vertebrate species. Rarefaction curves of host species identified via blood meal analysis suggest that the number of host diversity identified in *Ae. taeniorhynchus* is nearing an asymptote, while *Cx. nigripalpus* and *Cx. interrogator* likely interact with a broader range of hosts (Figure 5b).

Two human blood meals were identified in *Cx. erraticus* and *Cx. interrogator* from residential greenspaces, while three total human blood meals were found in *Ae. taeniorhynchus* (N = 2) and an unidentified mosquito (N = 1) in conservation greenspaces.

## 4. Discussion

This study aimed to characterize the mosquito assemblages and mosquito–host associations between two types of greenspaces found in Vero Beach, Florida. We observed distinct patterns of abundance for some mosquito species. *Aedes aegypti* and *Ae. albopictus*, important vectors of many pathogens, were significantly more common in the residential greenspaces than in conservation greenspaces. Residential greenspaces were surrounded by residential communities which could provide additional suitable larval habitats for these species. This finding supports previous findings that link these *Aedes* vectors with human-dominated environments [6,21,55].

In contrast, *Ae. taeniorhynchus* was significantly more abundant in the conservation greenspaces than in residential greenspaces. This pattern is likely due to the presence of *Ae. taeniorhynchus* larval habitats in our selected conservation greenspace sites which were located right along the edge of the Indian River Lagoon. *Aedes taeniorhynchus* is closely associated with salt marshes and coastal wetlands where it frequently emerges in large numbers. Therefore, its abundance likely reflects its preference for specific coastal habitats, such as salt marshes and wetlands, rather than a broad association with conservation greenspaces in general.

Although not an important vector species, *Ae. taeniorhynchus* poses a major biting nuisance to communities, prompting mosquito control programs in Florida to allocate substantial resources annually to manage its populations [56]. This species also showed the highest number of human blood meals in this study (*n* = 4), emphasizing its role as a biting nuisance. However, only seven human blood meals were detected in this study. Thus, our ability to draw broader conclusions regarding human and mosquito interactions is limited.

*Culex nigripalpus*, a vector for St. Louis encephalitis and West Nile virus (WNV) in Florida [57,58,59], was found in high numbers across both greenspace types, showing its broad habitat associations. Similarly, *Cx. quinquefasciatus* showed no significant difference in abundance between residential and conservation areas.

The rarefaction curve (Figure 2a) indicated similar species richness in mosquito assemblages across the two greenspace types. The high Jaccard Index further indicated a strong overlap in species between mosquito communities of both greenspace types. This similarity is likely influenced by the relatively short distances among sites and the proximity of conservation greenspaces to residential areas, despite being part of larger conservation lands.

We found significant climatic influences on mosquito abundance, measured by trap catches targeting host-seeking and resting females, across five species of interest (*Ae. taeniorhynchus*, *Ae. aegypti*, *Ae. albopictus*, *Cx. quinquefasciatus*, and *Cx. nigripalpus*). Generalized linear model analyses showed that temperature significantly affected the total abundance of all species except *Cx. quinquefasciatus*. This result was unexpected, as both high and low temperatures have been shown to negatively impact *Cx. quinquefasciatus* [60,61]. It is possible that the temperatures within the temperature range at our study sites did not markedly affect *Cx. quinquefasciatus*, a species generally well-adapted to warmer climates and shown to be more cold-tolerant than *Ae. aegypti* [40,41].

We found that accumulated rainfall three weeks prior to trapping significantly impacted the counts of *Cx. nigripalpus* and *Cx. quinquefasciatus*, but not the other three *Aedes* species. *Aedes taeniorhynchus* primarily breeds in brackish water found in salt marshes, with its population dynamics likely more influenced by tidal flooding than by rainfall [41,62]. This unique breeding habitat may make *Ae. taeniorhynchus* less sensitive to accumulated rainfall compared to other species, though previous studies show mixed results, with a study finding significant effects of rainfall on its abundance [63], while others do not [40,41]. Similarly, the literature on the relationship between rainfall and *Ae. aegypti* or *Ae. albopictus* abundance is inconsistent [44,64,65,66,67]. These species often breed in artificial water containers (e.g., birdbath, buckets, and flowerpot saucers), which can be maintained by household or gardening activities. This flexibility may reduce their dependence on recent rainfall patterns for breeding. In contrast, *Cx. nigripalpus* and *Cx. quinquefasciatus* are well adapted to temporary pools and standing water, which increase in availability after rainfall events. *Culex nigripalpus*, in particular, is a known floodwater species and exhibits seasonal abundance peaks closely linked to rainfall patterns in Florida [15,68,69].

In addition to analyzing mosquito community composition, we examined mosquito–host interactions through blood meal analysis and compared them across greenspace types. Habitat type can shape mosquito–host contact patterns in two primary ways. First, it determines which mosquito and vertebrate species are present, thus defining the possible mosquito–host pairs. Second, it affects the relative abundance of vertebrate hosts, which may further shape the blood meal composition of mosquitoes whose host ranges are broader. For example, Hancock and Camp investigated mosquito–host associations across urban, suburban, and rural habitats in Florida and found that *Cx. nigripalpus*, a species with a wide host range, fed significantly more on mammals in rural than in urban areas [70]. Faraji et al. compared blood feeding patterns of *Ae. albopictus* in suburban and urban areas of the northeastern USA and found that *Ae. albopictus* significantly fed on humans more frequently in suburban settings, while cat-derived blood meals were more common in urban areas [71]. The authors suggested that although human density was higher in urban areas, suburban residents spent more time outdoors, increasing their exposure to exophilic mosquitoes. Thus, the presence and abundance of mosquitoes in urban greenspaces may not directly translate disease risk for humans if the predominant mosquito species are not important vectors or, if they are, do not feed frequently on humans.

Statistical analyses were conducted on species that had sufficient samples with identified host sources. Our study found statistically significant host associations: *Cx. nigripalpus* fed more on birds, while *Ae. taeniorhynchus* fed more on mammals, generally consistent with the existing literature on their host association [72,73,74,75]. These host associations did not vary across our study sites, suggesting stable mosquito–host relationships for both species across the two greenspace types. However, as we did not survey vertebrate host availability within the study sites, it is not possible to determine whether the observed patterns are driven by host preferences or simply by host presence [25,76]. We also noted a high diversity of host species associated with *Cx. nigripalpus*. The widespread abundance of *Cx. nigripalpus* across all study sites, combined with its broader host range, emphasizes its potential role as a key vector for WNV in this area.

This study contains several limitations. First, the limited spatial distribution and small number of study sites prevents the generalizability of the observed differences in mosquito communities between conservation and residential greenspaces. The two coastal conservation greenspaces in this study, located adjacent to a lagoon, may not be representative of inland conservation greenspaces. Similarly, the two residential greenspaces, situated further from the lagoon in a relatively small city, may not reflect residential greenspaces elsewhere. Despite these limitations, our findings highlight that, under the conditions of these sites, the two types of greenspaces differ in their mosquito communities. This provides valuable insight into how greenspace types can influence mosquito communities differently, likely driven by variations in larval habitats within or near the greenspaces.

Second, this study’s design implemented LDA collection during dawn hours and CO_2_-baited traps during dusk-to-dawn hours. The mosquito collection was also restricted to four-week sampling intervals, followed by three weeks of no sampling. These periods of no sampling may have lost detailed mosquito dynamics, such as collection of mosquitoes active after dawn and before dusk hours or the impacts of weather-related variables.

Third, our study used weather records from an airport location, not at the study sites. Weather related variables, such as precipitation and temperature, may vary across sites and at smaller scales than our use of this nearby site could account for. Therefore, incorporating site-specific weather measurements could provide a more accurate understanding of the association between weather variables and mosquito dynamics.

Finally, blood meal analysis for this study was unable to determine mixed blood meals. During the chromatogram data analysis, three samples were suspected to have contained blood from multiple host species. In addition, due to DNA degradation, some blood-fed samples were lost. Of the 533 blood-fed mosquitoes collected, only 101 were successfully processed with their host species identified. This smaller data set from blood meal analysis limited our ability to draw statistically supported conclusions for many mosquito species.

## 5. Conclusions

This study provides insights into mosquito community composition and host associations in two greenspace types in a coastal city of Florida. In addition to reporting surveillance data on the mosquito community in these locations, we observed distinct patterns in the relative abundance of mosquito species. *Aedes taeniorhynchus*, an important biting nuisance but not a significant vector species, was more prevalent in preserved coastal wetland areas. In contrast, *Ae. aegypti* and *Ae. albopictus*, both key vector species, were more common in residential greenspaces. Blood meal analyses revealed stable host associations for *Cx. nigripalpus* and *Ae. taeniorhynchus*, indicating that greenspace type did not influence host feeding patterns for these species.

These findings highlight the importance of understanding mosquito ecology and host interactions in assessing potential mosquito-borne disease risks in urban landscapes. Our results indicate that greenspaces are not uniform in terms of their species’ relative abundances; certain greenspaces may pose a higher risk or bite nuisance due to the presence of specific species. Future studies could expand the research scale to further investigate how different types of greenspaces influence mosquito communities. This knowledge will help identify healthier landscape designs, emphasizing the potential for urban greenspace infrastructure to shape mosquito communities and impact public health. Including mosquito control considerations in the planning and reviewing process of new greenspace development and land use planning [77] could be beneficial for reducing mosquito-borne disease risks in urban areas.

## Figures and Tables

**Figure 1 insects-16-00233-f001:**
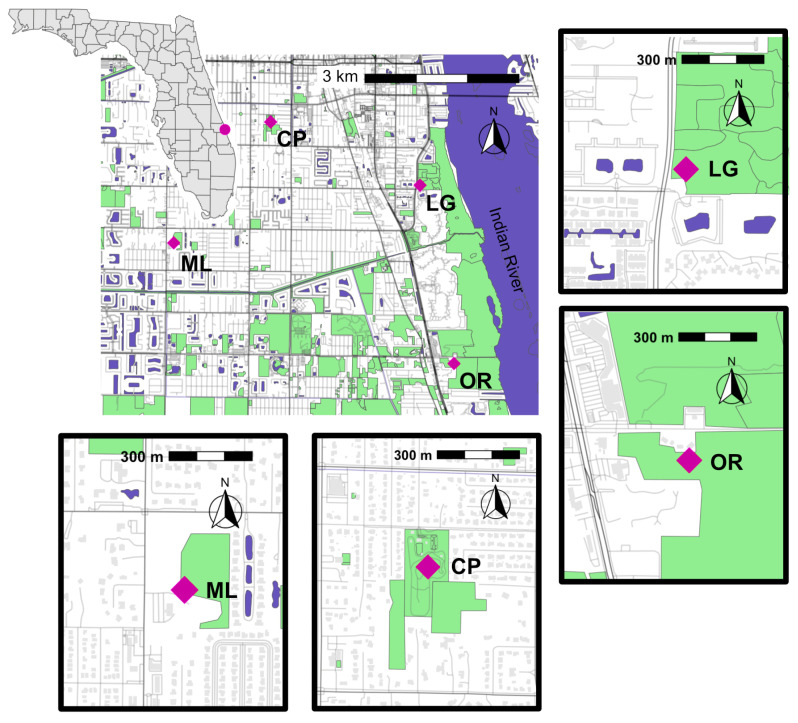
Maps showing locations of the four study sites located in the City of Vero Beach (circle dot on the (**top left**) Florida map). The study sites are marked by diamond shapes and include the following: Moose Lodge #1822 (ML; (**bottom left**)), Charles Park (CP; (**bottom middle**)), the Oslo Riverfront Conservation Area (OR; (**bottom right)),** and the Indian River Lagoon Greenway (LG; (**top right**)). The Indian River Lagoon Greenway and Oslo Riverfront Conservation Area sites represent conservation greenspaces, while Charles Park and Moose Lodge represent residential greenspaces.

**Figure 2 insects-16-00233-f002:**
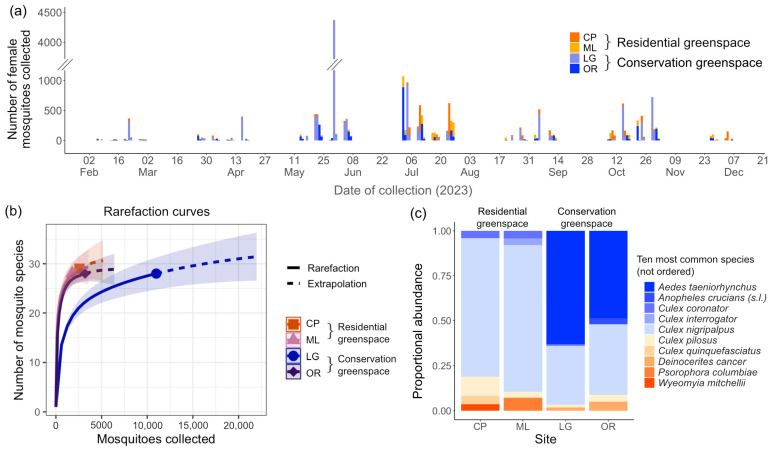
(**a**) Bar graph showing the total number of female mosquitoes collected for all sampling dates at different sites. (**b**) Rarefaction curves with 95% confidence intervals showing the expected number of mosquito species at different sampling efforts for each study site. (**c**) Stacked bar graph showing proportional abundance of the ten most common species across all study sites. Study site names are abbreviated as CP (Charles Park), ML (Moose Lodge), LG (Indian River Lagoon Greenway), and OR (Oslo Riverfront Conservation Area).

**Figure 3 insects-16-00233-f003:**
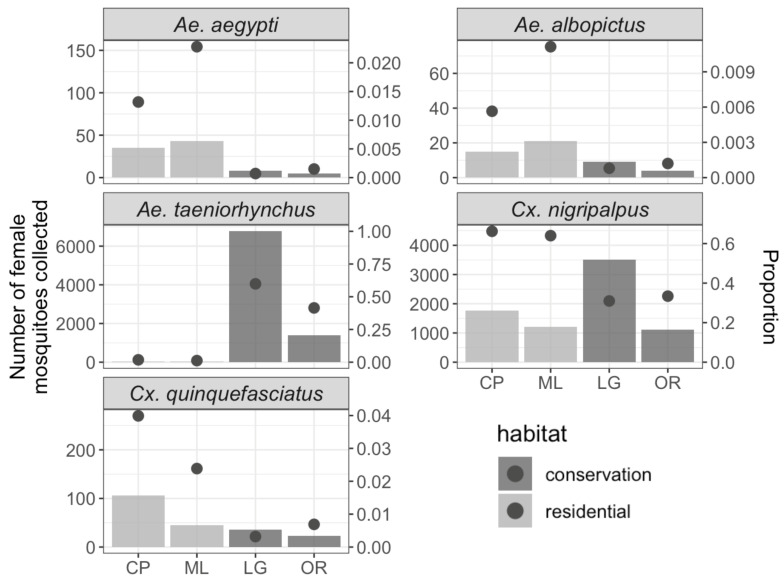
Graphs showing the total number of female mosquitoes (bars, (**left**) *y*-axis) and their proportion (dots, (**right**) *y*-axis) of five key species: *Ae. aegypti*, *Ae albopictus*, *Ae. taeniorhynchus*, *Cx. nigripalpus*, and *Cx. quinquefasciatus*. Study sites listed on the *x*-axis include Charles Park (CP) and Moose Lodge (ML) which represented residential greenspaces, and the Indian River Lagoon Greenway (LG) and the Oslo Riverfront Conservation Area (OR) which represented conservation greenspaces. The *y*-axis scale differs across species panels to account for differences in the total number and proportion of female mosquitoes collected for each species.

**Figure 4 insects-16-00233-f004:**
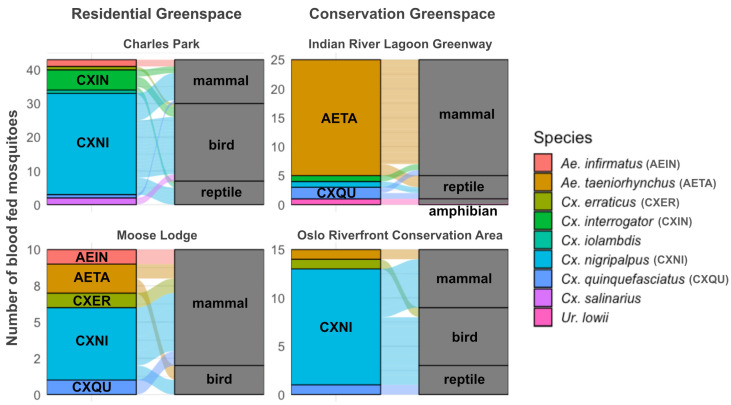
An alluvial diagram showing the relationship between mosquito species (**left bar**) and host type (**right bar**) across four study sites based on blood meal analysis. The *Y*-axis shows the number of blood-fed mosquitoes. Each mosquito species is labeled with an abbreviated name, when space allowed, and colors. Flowlines between the bars indicate mosquito–host interactions via blood feeding with thickness of the flowlines indicating the relative frequency of each host type used by the mosquito species.

**Figure 5 insects-16-00233-f005:**
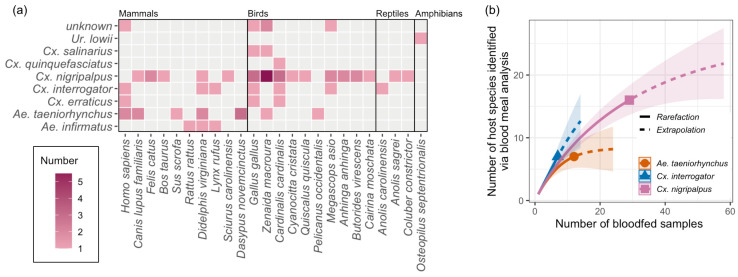
(**a**) Heat map showing interactions between mosquito species and host species via blood feeding. Note: *Rattus rattus* L. and *Rattus norvegicus* Berkenhout cannot be distinguished using the *COI* gene sequence, and both rodent species are sympatric in Florida. (**b**) Rarefaction curve showing the expected number of host species identified via blood meal analysis as a function of the number of blood-fed samples. This analysis focuses on the three mosquito species for which the blood hosts of at least seven blood-fed females were identified to the species level.

**Table 1 insects-16-00233-t001:** Number of species identified per site (species richness), along with the Chao1, ACE, Shannon, Simpson, and Pielou diversity indices listed for each site.

	Residential Greenspace		Conservation Greenspace	
	Charles Park	Moose Lodge	Indian River Lagoon Greenway	Oslo Riverfront Conservation Area
Species Richness	29	28	28	28
Chao1	30.00	28.75	31.33	29.00
ACE	30.98	29.99	32.28	28.71
Simpson Diversity Index	0.52	0.53	0.52	0.69
Shannon Diversity Index	1.42	1.53	0.99	1.64
Pielou Evenness Index	0.34	0.33	0.37	0.32

**Table 2 insects-16-00233-t002:** Results from negative binomial generalized linear regression model analyses examining associations between weather variables of interest, greenspace type, and counts of female mosquitoes belonging to five key species. Temperature and rainfall were measured in °F and inch, respectively. Data for all models included 103 observations.

Species	Variable	Estimate	Std. Error	Z Value	*p*-Value
*Ae. taeniorhynchus* (Pseudo-R^2^ = 0.51)	Intercept	0.7726	0.1592	4.851	<0.001
	Total rain 21 days prior	−0.0008	0.0367	−0.022	0.982
	Temperature (1st polynomial)	6.2496	1.5594	4.008	<0.001
	Temperature (2nd polynomial)	−2.9715	1.4662	−2.027	0.043
	Autoregressive term	0.0002	0.0001	2.179	0.029
	Greenspace type (residential)	−1.8681	0.2451	−7.621	<0.001
*Cx. nigripalpus* (Pseudo-R^2^ = 0.23)	Intercept	3.0497	0.2825	10.795	<0.001
	Total rain 21 days prior	0.2929	0.0593	4.932	<0.001
	Temperature (1st polynomial)	2.5252	1.4726	1.715	0.086
	Temperature (2nd polynomial)	−4.8226	1.5928	−3.028	0.003
	Autoregressive term	0.0037	0.0011	3.258	0.001
	Greenspace type (residential)	−0.5148	0.278	−1.852	0.064
*Ae. aegypti* (Pseudo-R^2^ = 0.38)	Intercept	−12.2938	2.7787	−4.424	<0.001
	Total rain 21 days prior	0.0269	0.0706	0.381	0.703
	Temperature	0.1362	0.0352	3.874	<0.001
	Greenspace type (residential)	1.911	0.4092	4.67	<0.001
*Ae. albopictus* (Pseudo-R^2^ = 0.21)	Intercept	−9.9562	2.9866	−3.334	<0.001
	Total rain 21 days prior	0.0877	0.0764	1.148	0.251
	Temperature	0.1052	0.038	2.766	0.006
	Greenspace type (residential)	1.0006	0.4278	2.339	0.019
*Cx. quinquefasciatus* (Pseudo-R^2^ = 0.13)	Intercept	−2.2039	2.2756	−0.969	0.333
	Total rain 21 days prior	0.1898	0.0819	2.316	0.021
	Temperature	0.0228	0.03	0.76	0.447
	Greenspace type (residential)	0.6871	0.418	1.644	0.100

## Data Availability

The original data contributions presented in this study are included in this article/Appendix A. Further inquiries can be directed to the corresponding author.

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
