# Peer review of "Mosquito Population Dynamics and Blood Host Associations in Two Types of Urban Greenspaces in Coastal Florida"

_insects, 2025, doi:10.3390/insects16030233_

Round 1
Reviewer 1 Report
Comments and Suggestions for Authors
Generally well presented and nice paper, could be simplified in areas to present the message a bit more succinctly but otherwise reads well.
Notes:
- needs more stats in the text with p values shown after different statements given, see lines 363-391 for example.
- A little overwritten and maybe misses the feeling of reporting / surveillance, since the study is more of a surveillance dataset than a unique or novel study. This data is fine but it might be trying to say more than there actually is to say.
- Bloodmeal vs blood meal, also bloodfed vs blood fed. please stick to one format (bloodmeal is my preference).
Lines 36 - 39 - perhaps highlight anything specific you can ID here, because these host preferences are already well documented.
Lines 104-119 - reads more like discussion, might fit better there if not there already?
Line 120 - I am a little shocked that this is true, given how many mosquito papers exist, and the data may simply be buried in reports. E.g. see https://bioone.org/journals/Journal-of-the-American-Mosquito-Control-Association/volume-30/issue-2/13-6388.1/A-Taxonomic-Checklist-of-the-Mosquitoes-of-Iowa/10.2987/13-6388.1.short
Line 161-163 - details should be in the materials and methods
Line 185-186 - formatting between sections - check for indentations also for headings
Line 203 - Can you please include primer sequences here e.g. (5'-GAGTTTCA-3') or in a table, can go into the supplement. If access to cited references are lost or not accessible, the reader can't check the actual sequences ordered and used for the study.
Line 207 - 16 out of how many? Specify here i.e. 16/XX
Line 289 - sypnumber please expand on this.
line 297 - can you provide a stat on this, and per site, chisq or similar?
Figure 2a- would be nice to see weather data above to see relation to the collection, but the figure is well made.
Figure 2c - please provide full species names in figure since many are not mentioned in the text beforehand.
Figure 3 - should be shown as proportion of mosquitoes collected at each site to normalise these numbers.
Line 356-361 - what does this mean? Please write more clearly about what it signifies.
Table 2 - should be in supplement while the actual model results should be in the main text.
Line 421 - Culex salinarius needs italics.
Author Response
- Comment: needs more stats in the text with p values shown after different statements given, see lines 363-391 for example.
Response: We sincerely thank the reviewer for their time and effort in improving our manuscript. We have carefully reviewed the Results section to ensure that all appropriate statistics and p-values are reported when statements indicating statistical significance are indicated. For the section referenced here (originally lines 363–391), we have included only the p-values for brevity, as the full statistical details are now provided in the revised Table 2.
- Comment: A little overwritten and maybe misses the feeling of reporting / surveillance, since the study is more of a surveillance dataset than a unique or novel study. This data is fine but it might be trying to say more than there actually is to say.
Response: We appreciate the reviewer’s feedback and recognize the importance of framing this study as a surveillance dataset. We have placed greater emphasis on the surveillance and reporting aspects of our study in the revised version. Specifically, we have added concise statements in the Simple Summary, Abstract, Introduction, and Summary sections to highlight that the study reports mosquito surveillance data from Vero Beach. For example, in the Abstract, we now state: "This study reports mosquito surveillance and bloodmeal analysis data from greenspaces in Vero Beach, Florida. It also compares mosquito assemblages and host associations across residential and conservation greenspaces to assess how greenspace type impacts mosquito abundance and host interactions."
- Comment: Bloodmeal vs blood meal, also bloodfed vs blood fed. please stick to one format (bloodmeal is my preference).
Response: We appreciate the reviewer’s feedback and revised the manuscript to ensure consistency in our word choice. We opted to use “blood meal” and “blood fed” to follow the word choice of the Reeves et al. 2018 publication.
- Comment: Lines 36 - 39 - perhaps highlight anything specific you can ID here, because these host preferences are already well documented.
Response: To meet Insects’ 200-word abstract limit, we have removed this statement (originally lines 36–39). As the reviewer noted, these host preferences are well documented, and we determined that their removal would not impact the overall clarity or significance of the abstract.
- Comment: Lines 104-119 - reads more like discussion, might fit better there if not there already?
Response: We agree with the reviewer. We have revised and moved this paragraph to the Discussion section.
- Comment: Line 120 - I am a little shocked that this is true, given how many mosquito papers exist, and the data may simply be buried in reports. E.g. see https://bioone.org/journals/Journal-of-the-American-Mosquito-Control-Association/volume-30/issue-2/13-6388.1/A-Taxonomic-Checklist-of-the-Mosquitoes-of-Iowa/10.2987/13-6388.1.short
Response: We agree that mosquito surveillance data from various greenspaces may already exist in the literature, including reports such as the one suggested by the reviewer. However, relatively few studies exist that were designed to directly compare mosquito communities across different types of urban greenspaces. Our search yielded limited studies of this nature. This gap in knowledge was also highlighted in a recent meta-analysis on this topic (Rhodes et al. 2022, https://pubmed.ncbi.nlm.nih.gov/35323569/). That said, since the statement in question is somewhat subjective and not central to our study, we have decided to remove it for clarity.
- Comment: Line 161-163 - details should be in the materials and methods
Response: We have now removed the map creation details from the figure description and added to the materials and methods text, Study Site section.
- Comment: Line 185-186 - formatting between sections - check for indentations also for headings.
Response: The formatting of section titles and section title indentations have been adjusted to follow the Insects manuscript template.
- Comment: Line 203 - Can you please include primer sequences here e.g. (5'-GAGTTTCA-3') or in a table, can go into the supplement. If access to cited references are lost or not accessible, the reader can't check the actual sequences ordered and used for the study.
Response: The primer sequences listed in the Reeves et al. 2018 publication and used for this study have now been included as Supplemental Data 1).
- Comment: Line 207 - 16 out of how many? Specify here i.e. 16/XX.
Response: This statement has been clarified to explain that there were 245 unsuccessfully amplified samples and using the other available primer combinations, 16 samples were later successfully amplified.
- Comment: Line 289 - sypnumber please expand on this.
Response: This was a typing error for the word “number”.
- Comment: line 297 - can you provide a stat on this, and per site, chisq or similar?
Response: Thank you for the suggestion. We have included chi-square goodness-of-fit tests to compare mosquito abundances among the four sites as well as between the conservation and residential habitats.
- Comment: Figure 2a- would be nice to see weather data above to see relation to the collection, but the figure is well made.
Response: Thank you for the suggestion. Due to scale constraints, integrating all weather data into the bar graph while maintaining clarity and appropriate chart size for the main manuscript was challenging. We decided to include the requested figure as Supplementary Data 5, where the relationship between weather patterns and mosquito abundance cane be more clearly visualized.
- Comment: Figure 2c - please provide full species names in figure since many are not mentioned in the text beforehand.
Response: Figure 2c now includes full species names in the figure legend.
- Comment: Figure 3 - should be shown as proportion of mosquitoes collected at each site to normalize these numbers.
Response: Our objective in including Figure 3 was to focus on the differences in the raw counts of important vector species across sites, in addition to the proportions of common species already presented in Figure 2c. Since trapping effort was standardized across all sites, comparing raw count data can provide additional valuable insights into potential bite exposure risk from vectors. However, we agree that including proportions in Figure 3 would offer a more comprehensive view. To address this, we have revised Figure 3 to display both total counts (bars) and proportions (dots). In some cases, the trends are consistent between raw counts and proportions (e.g., Aedes aegypti and Aedes taeniorhynchus; see revised Figure 3). However, for Culex nigripalpus: there was no clear pattern in raw counts, whereas proportions were higher in residential sites than in conservation sites. This difference points to the importance of presenting both metrics to fully capture site-specific differences.
- Comment: Line 356-361 - what does this mean? Please write more clearly about what it signifies.
Response: We have revised the paragraph to explicitly explain that the Jaccard similarity index measures species overlap between two communities. We also noted that a Jaccard value of 1 indicates the communities share exactly the same species.
- Comment: Table 2 - should be in supplement while the actual model results should be in the main text.
Response: Table 2 has been revised to show detailed result of the models, listing variable estimates, standard errors, z-, and p-values, as well as the overall models’ pseudo-R2. The supplementary data 3 now included additional model statistics and model evaluation reports.
- Comment: Line 421 - Culex salinarius needs italics.
Response: The text font has been italicized.
Reviewer 2 Report
Comments and Suggestions for Authors
1. The address labeled as number 3 is missing when listing the authors' addresses.
2. If possible, please list the keywords in alphabetical order.
3. According to the author guidelines of the journal Insects, references cited both in the text and in the references section must be numbered sequentially.
4. All species and genus names used in the text and references must be written in italics. The first time a species name appears in the text, it should be presented together with the name of the author who originally described it. For subsequent mentions, use the abbreviated species or genus name. Culex as Cx.; Aedes as Ae. Lines 93, 126, 290, 421, 494, 542, 648 etc.
5. Including climatograms/climate diagrams showing the precipitation and temperature values of the study areas for research period could provide meaningful context for the research.
6. Was only one COâ‚‚-baited or light trap used per sampling site or more? What was the distance between the traps? What specific factors were considered when setting up the traps? For example, the height above the ground, distance from humans or animals, proximity to or distance from vegetation, etc.
Comments on the Quality of English LanguageGood
Author Response
- Comment: The address labeled as number 3 is missing when listing the authors' addresses.
Response: We thank the reviewer for their time and effort in helping to improve our manuscript. The address labeled as number 3 on the title page is now present for all co-authors who are affiliated with that location.
- Comment: If possible, please list the keywords in alphabetical order.
Response: The keywords have been rearranged into alphabetical order.
- Comment: According to the author guidelines of the journal Insects, references cited both in the text and in the references section must be numbered sequentially.
Response: The references have been rearranged to follow the journal reference preference, in numerical order of appearance.
- Comment: All species and genus names used in the text and references must be written in italics. The first time a species name appears in the text, it should be presented together with the name of the author who originally described it. For subsequent mentions, use the abbreviated species or genus name. Culex as Cx.; Aedes as Ae. Lines 93, 126, 290, 421, 494, 542, 648 etc.
Response: We have revised our manuscript as suggested. All species appearing for the first time, in-text, now have the name of the original describer listed. All following species reference use the abbreviated species name, unless it is used at the beginning of a sentence.
- Comment: Including climatograms/climate diagrams showing the precipitation and temperature values of the study areas for research period could provide meaningful context for the research.
Response: Thank you for the suggestion. Similarly, Reviewer 1 also suggested to include a plot of precipitation/temperature values along with mosquito abundance in Figure 2a. However, due to scale constraints, integrating all weather data into the bar graph while maintaining clarity and appropriate chart size for the main manuscript was challenging. We decided to include the requested figure as Supplementary Data 5, where the relationship between weather patterns and mosquito abundance is more clearly visualized.
- Comment: Was only one COâ‚‚-baited or light trap used per sampling site or more? What was the distance between the traps? What specific factors were considered when setting up the traps? For example, the height above the ground, distance from humans or animals, proximity to or distance from vegetation, etc.
Response: Two COâ‚‚-baited traps were used per sampling site and were hung at least 0.32km from each other in opposing directions. The COâ‚‚-baited traps were hung at least 0.61m above the ground and no more than 1.22m off the ground. The COâ‚‚-baited traps at each site were near or directly above brush/vegetation, as well small collections of water (e.g., partially filled ditches, water-filled containers, etc.) when available. The distance between the COâ‚‚-baited traps and nearby vegetation or residential buildings was not measured for each trapping session. We have included these details in the revised manuscript.